# Mineralogical Fingerprint of Iron Ore Tailings in Paraopeba River Bedload Sediments after the B1 Dam Failure in Brumadinho, MG (Brazil)

**Fernando Verassani Laureano** [1]**, Rogerio Kwitko-Ribeiro** [2,*]**, Lorena Guimarães** [2] **and Lucas Pereira Leão** [3,*]

1   Vale S.A., Diretoria de Reparação, Nova Lima 34006-049, MG, Brazil; fernando.laureano@vale.com
2   Vale S.A., Centro de Desenvolvimento Mineral (CDM), Santa Luzia 33040-900, MG, Brazil; lorena.guimaraes@vale.com
3   Departamento de Geologia, Escola de Minas, Universidade Federal de Ouro Preto, Ouro Preto 35400-000, MG, Brazil
*   Correspondence: rogerio.kwitko@vale.com (R.K.-R.); lucas.leao@ufop.edu.br (L.P.L.)

**Abstract:** The study presents SEM-based automated mineralogy to distinguish between natural sediments and iron ore tailings deposits from the Paraopeba River, after the failure of B1 Dam in Brumadinho, Minas Gerais, Brazil. Samples were obtained from borehole cores drilled over channel bars and banks eight months after the failure. After preliminary facies description, sediments from 54 chosen intervals were subjected to density measurement, X-ray diffraction (XRD), SEM-based automated mineralogy (QEMSCAN) analysis and determination of geochemical major components. Hierarchical clustering analysis (HCA) and principal component analysis (PCA) revealed six main mineral associations governed by different contents and ratios of quartz, kaolinite and hematite. Natural sediments are predominantly composed of mineral associations containing kaolinite, quartz and quartz + hematite with density values ranging from 2.5 to 3.3 g/cm$^3$. Tailings deposits have density values higher than 3.5 g/cm$^3$ and are mainly composed of hematite with occasional occurrences of kaolinite + hematite. Because of geological complexity and historical terrain occupation and usage, geochemical anomalies are common in the Paraopeba River sediments. Our data suggests that mineralogical oriented studies should precede detailed geochemical investigations, to enhance the understanding of the source of such anomalies and the environmental jeopardy associated to the occurrence. In this sense, SEM-based mineralogy has an enormous potential in environment studies.

**Keywords:** B1 Dam failure; Paraopeba river sediment; tailings; mineralogy

## 1. Introduction

Mining tailings are currently amongst the largest volumes of waste material worldwide [1]. Most of these materials have been historically stored in tailings impoundment areas behind dams, especially when wet ore treatment is required. Tailings impoundment areas are a necessary part of the ore treatment process, but their failure can have some of the largest negative environmental consequences associated with mining activity [2]. Dam burst events have doubled in the past 20 years [3] and caused harmful impacts in many countries, including Brazil, Mexico, Canada, Philippines, Finland, Hungary, Russia and China [4].

River sediments are one of the environmental zones most compromised by tailing dam failure, due in part to sediment absorption/adsorption capacity, that characterizes them as large geochemical deposits [5]. The deposition of extensive waste mass in water bodies can increase the concentration of metal and other toxic substances in sediments which, in turn, triggers environmental quality loss and threatens the fluvial ecosystem [6,7]. On the other hand, sediments function as natural records for recent environmental changes [8] and can provide important proxies for measuring the distribution patterns of contaminants in aqueous systems [9–13].

However, addressing the provenance of potentially toxic elements can be challenging, especially when contaminants spread over watersheds with a complex geological background and intense human activities [14,15]. In this sense, mineralogical and textural characterization of the natural sediments is fundamental for further quantitative analysis and assessment of the degree of impacts generated by tailings contamination.

On 25 January 2019 the B1 Dam collapsed in the city of Brumadinho, Minas Gerais, Brazil. Reaching 86 m high, it was the main tailing impoundment from an iron ore mining complex named Mina do Córrego do Feijão, whose operation started in the 1960s. According to recent estimates [16], the total waste overflown volume was about 9.7 Mm$^3$. Approximately 8.1 Mm$^3$ was deposited in the Ferro-Carvão creek valley (Figure 1), and 1.6 Mm$^3$ settled into the Paraopeba river, resulting in a direct impact to the channel sediment composition and dynamics. In addition to the socio-economic impact and those associated with the fauna and flora, studies suggest that the water quality and fluvial sediments in the Paraopeba river basin were also affected (i.e., [17–21]).

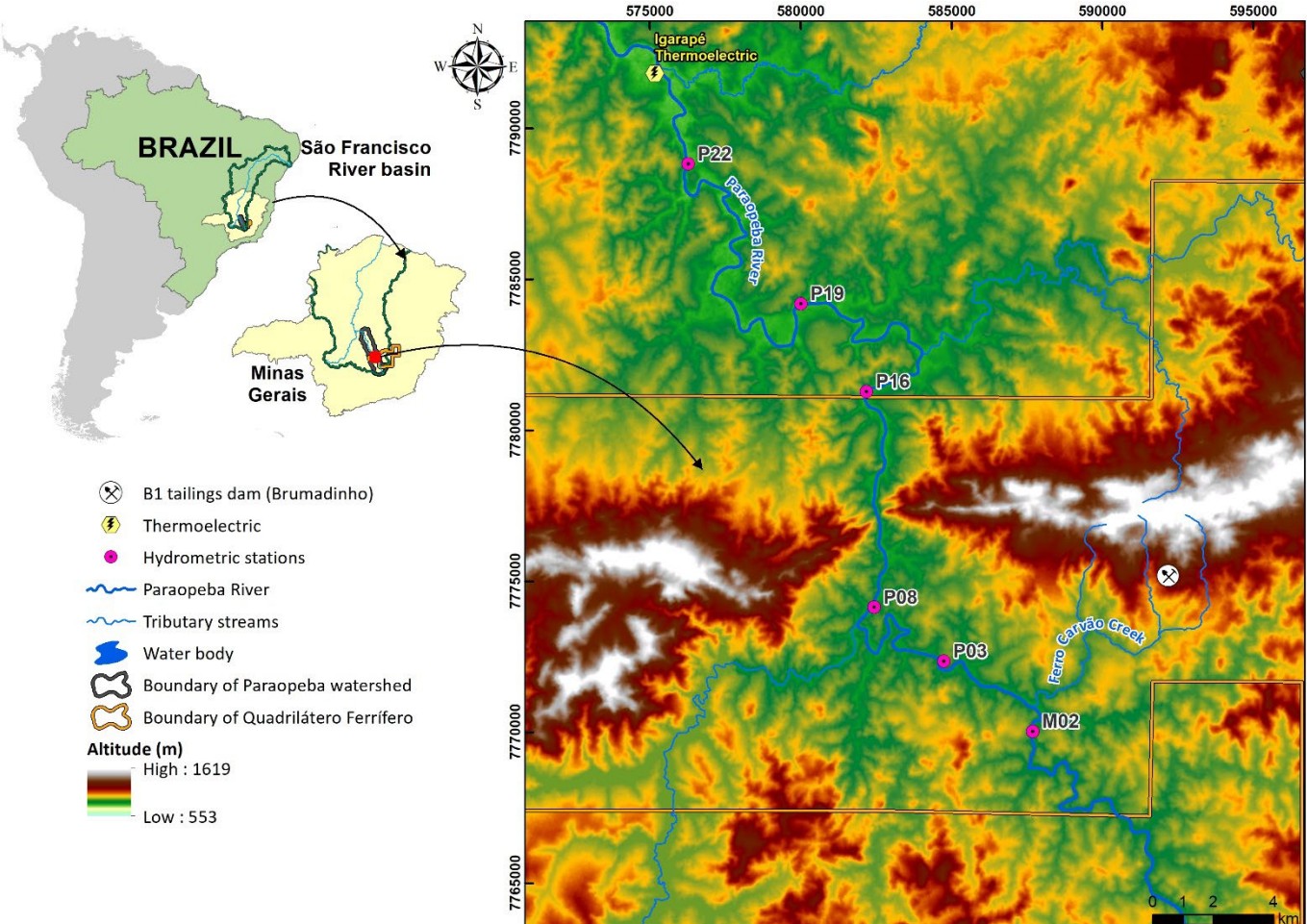

**Figure 1.** Hypsometric framework of Paraopeba River basin in the study area (Minas Gerais, Brazil).

In this paper, we present multiproxy data, including SEM-based automated mineralogy, to distinguish natural Paraopeba River sediments from those affected by tailings and other materials derived from B1 Dam collapse. Our results suggest that mineral content is a key factor for the distinction of these materials and can be used as a fingerprint for tailings presence in the channel sediments. Results are also used to discuss the importance of mineralogical studies prior to geochemical investigations when assessing sediments affected by tailings dams failure, especially when stratigraphic control is not properly achieved.

## 2. Study Area

The Paraopeba river is a right-hand tributary of Sao Francisco River, an important water source in Brazilian interior lands; draining the states of Minas Gerais, Bahia, Pernambuco, and Sergipe. The Paraopeba watershed has an elongated form, extending approximately 537 km long (Figure 1). The catchment area is 12,091 km$^2$, encompassing 48 municipalities with an estimated population of 1.318.885 people [22]. The hydrographic basin is divided into High, Medium and Low Paraopeba, and because those sectors are distinguished by elevation values, they also reflect the general topography, that ranges from mountainous to smooth relief [23,24] (Figure 1).

The east border of the watershed partially follows the highlands and ranges of an important Brazilian mineral province, named the Quadrilátero Ferrífero (Iron Quadrangle). Settlement of the region started at the end of the 17th century driven by gold discoveries, mainly in the High and Medium Paraopeba streams. The São Francisco and Paraopeba rivers were used as trade routes, which favored farming development [25]. Nowadays the major economic enterprises are mining, metallurgy and farming. The main water uses aside from human consumption are mining, crop irrigation and power generation [26].

The Paraopeba watershed displays a complex geological and geochemical framework, generally associated with Precambrian rocks. The Archean Rio das Velhas greenstone belt, composed of mafic and ultramafic rocks, is distributed along the High Paraopeba sector [27]; Paleoproterozoic metasedimentary units such as quartzites, phyllites, dolomites and itabirites (banded quartz-hematite iron formation) of the Minas Supergroup outcrop in the High and Medium sectors [28]; Neoproterozoic sedimentary cover makes up the substrate in the Low sector (São Francisco Supergroup, Bambuí Group) [29]. Crystalline basement composed of polydeformed gneissic rocks of tonalitic to granitic composition occur throughout the whole basin [30].

Important Fe reserves are concentrated within the itabirites of the Itabira Group (Minas Supergroup), whose genesis have been attributed to Superior Lake deposits [31]. Mn ore also occurs in association with itabirites, forming iron-manganese ores [32].

The mineral content of the sediments from the Paraopeba River reflectweathering of the outcropping rocks from the entire catchment. The predominance of quartz, kaolinite and clay-minerals reflects granite-gneissic, quartz and iron oxide complexes in the Minas Supergroup [33], while gibbsite and clay-minerals are likely sourced from the Bambuí Group [34].

Geology and historical occupation need to be considered together when assessing sediment geochemical composition in the Paraopeba river [15]. The influence of the former factor led Vicq et al. (2015) [35] to suggest specific reference values for sediment quality assessment in rivers of the Quadrilátero Ferrífero. On the other hand, it is undeniable that 300 years of human activities have played a role on the modern total load that flows into the Paraopeba River. Regarding B1 Dam tailings, it is important to mention that during the life of the Córrego do Feijão Mine no metallurgical ore treatment was performed except those physical processes necessary to achieve particle comminution. The tailings mineral assemblage was primarily hematite and quartz, and secondarily kaolinite and gibbsite [36]. The mean chemical composition calculated by this author was given by iron oxides (48%) and silica (20.6%), followed by aluminum and manganese oxides (2.5% and 1.0%, respectively).

## 3. Materials and Methods

### 3.1. Sampling

Samples were collected from six cores identified as M2, P3, P8, P16, P19 and P22 obtained from a drilling program executed over bars and channel banks of the rio Paropeba during 2019 September (end of dry season). Boreholes were located upstream from the confluence of Ferro-Carvão creek and Paraopeba river until the spillway of the Igarapé Thermoelectric Plant (Figure 1). The drilling equipment used was a stainless steel vibracoring system, 5 m long and 3 inches in diameter. At the base, a catcher device minimizes

material loss during core extraction, especially the finer particles such as clay and silt. PVC liners were used within the stainless steel rod as referenced in the Brazilian guide for sediment sampling and samples conservation [37].

### 3.2. Analytical Procedures

### 3.2.1. Macroscopic Description

The sediment cores were exposed in two half gutters so structural and textural descriptions could be carried out. Sedimentary structures, unconformities, grain size, sorting and rounding were recorded by visual and tactile means. For particles larger than silt, grain size was visually estimated, aided by auxiliary classification sheets [38], following the Wentworth division [39]. Sorting and rounding degrees for sand particles were described by visual and microscopic analyses, also aided by classification auxiliary sheets [40,41]. The content of opaque minerals was visually estimated as well. For sand particles, the mineral proportion was estimated with auxiliary sheets of Terry and Chilingar (1955) [42].

### 3.2.2. Sampling and Physicochemical Characterization

Samples were collected in order to represent the different strata observed in the six cores. In total, 54 samples were obtained along the length of each core as follows: M2 ($n = 10$), P3 ($n = 07$), P8 ($n = 10$), P16 ($n = 08$), P19 ($n = 10$) and P22 ($n = 09$).

The true density of samples was measured by a helium gas displacement ultra-pycnometer (Quantachrome Instruments ULTRAPYC 1200e, Boynton Beach, FL, USA). The chemistry of major elements (Ca, Fe, Ti, Mn, P, Al, Mg and Si) was determined by lithium borate fused disc and X-ray fluorescence (XRF Panalytical Axios Minerals), after calcination at 1000 °C.

### 3.2.3. Mineralogy

The samples were analyzed by X-ray diffraction Bruker D8 Advance, with 35 kV and 40 mA conditions (XRD). Results were subjected to hierarchical cluster analysis (HCA) and principal component analysis (PCA). HCA is a technique of multivariable association for gathering similar data in separated classes, ranked by likelihood. In order to have the data clustered with minimal loss of original information and mineralogical meaningfulness, a cut-off was set to 85 in the dendrogram of the 54 samples, allowing the recognition of six groups of samples.

For the SEM-based techniques, polished sections were produced following a proprietary method of cold epoxy embedding under centrifugation [43]. The preparation with sample to epoxy ratio of 1:2 (vol) is centrifuged, demolded, cut along the vertical axis and potted again in a 30 mm round mold to ensure the representativeness of a single surface in terms of morphology, size, density and particle composition. The modal mineralogy was generated using a QEMSCAN® system for SEM-based automated mineralogy, consisting of a FEI Quanta 650 SEM with two Bruker XFlash 6 | 30 EDS detectors. All measurements were produced in the Field Image mode, under 25 kV of beam acceleration and 10 nA of sample current, using 10 μm × 10 μm pixel spacing grid and 1500 X-ray counts per pixel. An average of 3.8 million pixels per sample were collected, over 30% of which were identified as mineral phases. For backscattered electron (BSE) imaging, a Hitachi SU3500 SEM was used, operating at 25 kV of beam acceleration.

Assay reconciliation is an important data quality check used for the validation of mineralogical data. It is completed by comparing a QEMSCAN calculated chemical assay (obtained from mineral chemistry with externally measured chemical data and the modal mineralogy result) with a tradition chemical assay by XRF or ICP-OES. The data are plotted against one another on the chart for visual comparison and identification of potentially anomalous measurements. A slight dispersion from 1:1 regression, as observed in the Figure 2, is expected and is due to fluctuations in mineral chemistry from an average stoichiometry.

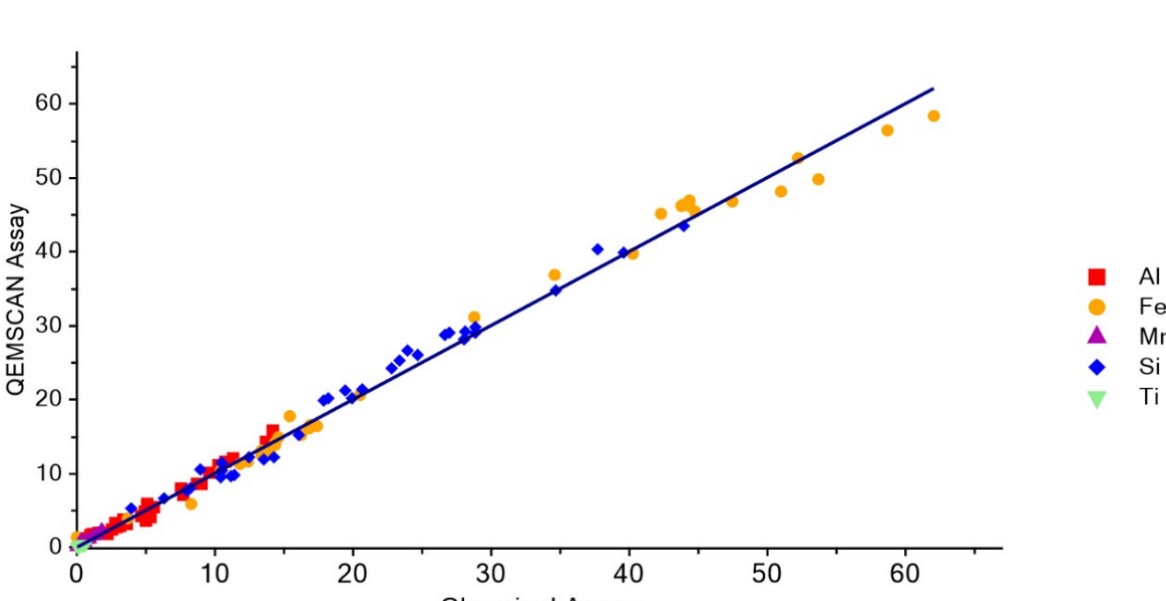

**Figure 2.** Assay reconciliation of QEMSCAN measurements.

## 4. Results

Textural descriptions of the 6 cores from the top to the bottom are individually described below. Selected sections from each one can be directly compared in the Figure 3. All boreholes are subject to the presence of tailings except for M2, which was located upstream of the confluence between Ferro-Carvão creek and Paraopeba river, (see Figure 1).

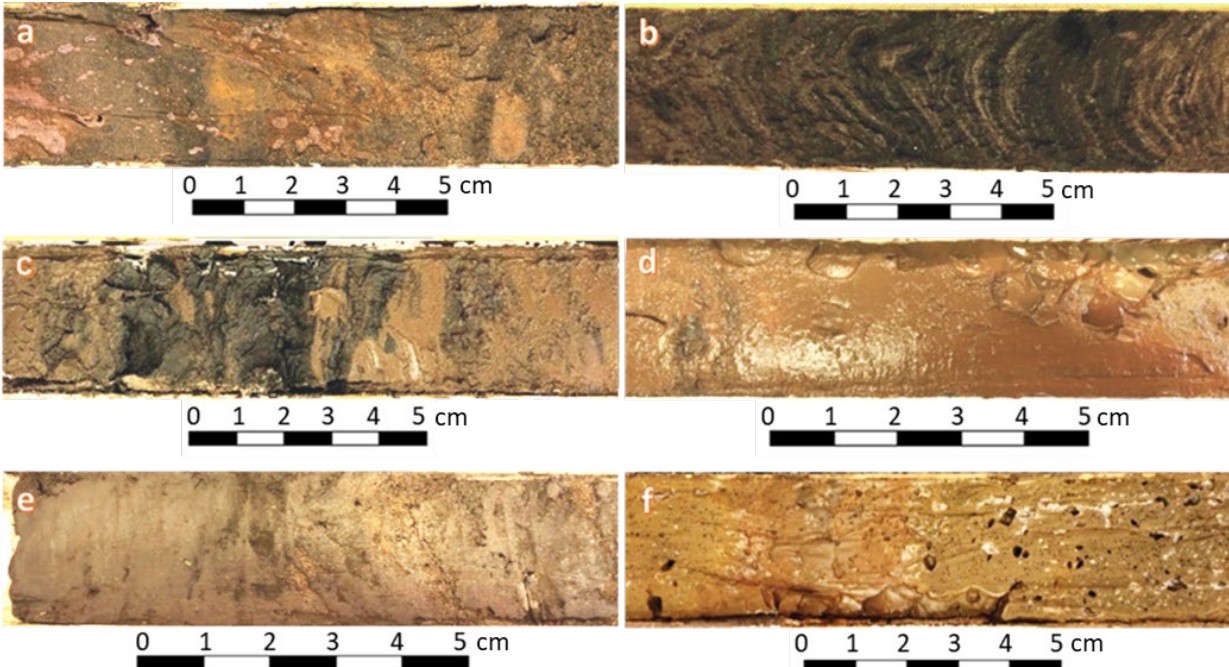

**Figure 3.** Photography of part of the sampled cores: (**a**)—M2 section; (**b**)—P3 section; (**c**)—P8 section; (**d**)—P16 section; (**e**)—P19 section; (**f**)—P22 section.

Figure 4 presents the results of SEM imaging by Back Scattered Electrons (BSE), making it is possible to identify the main minerals associated in each sample. In sample

M2-01 (Figure 4A), kaolinite and quartz stand out and few occurrences of iron oxides and hydroxides. Martite and goehtite with grains greater than 500 μm prevail in sample P3-02 (Figure 4B). Sample P8-01 (Figure 4C) is composed of quartz, iron oxides and hydroxides that are associated with a very fine matrix, making it impossible to identify the associated minerals.

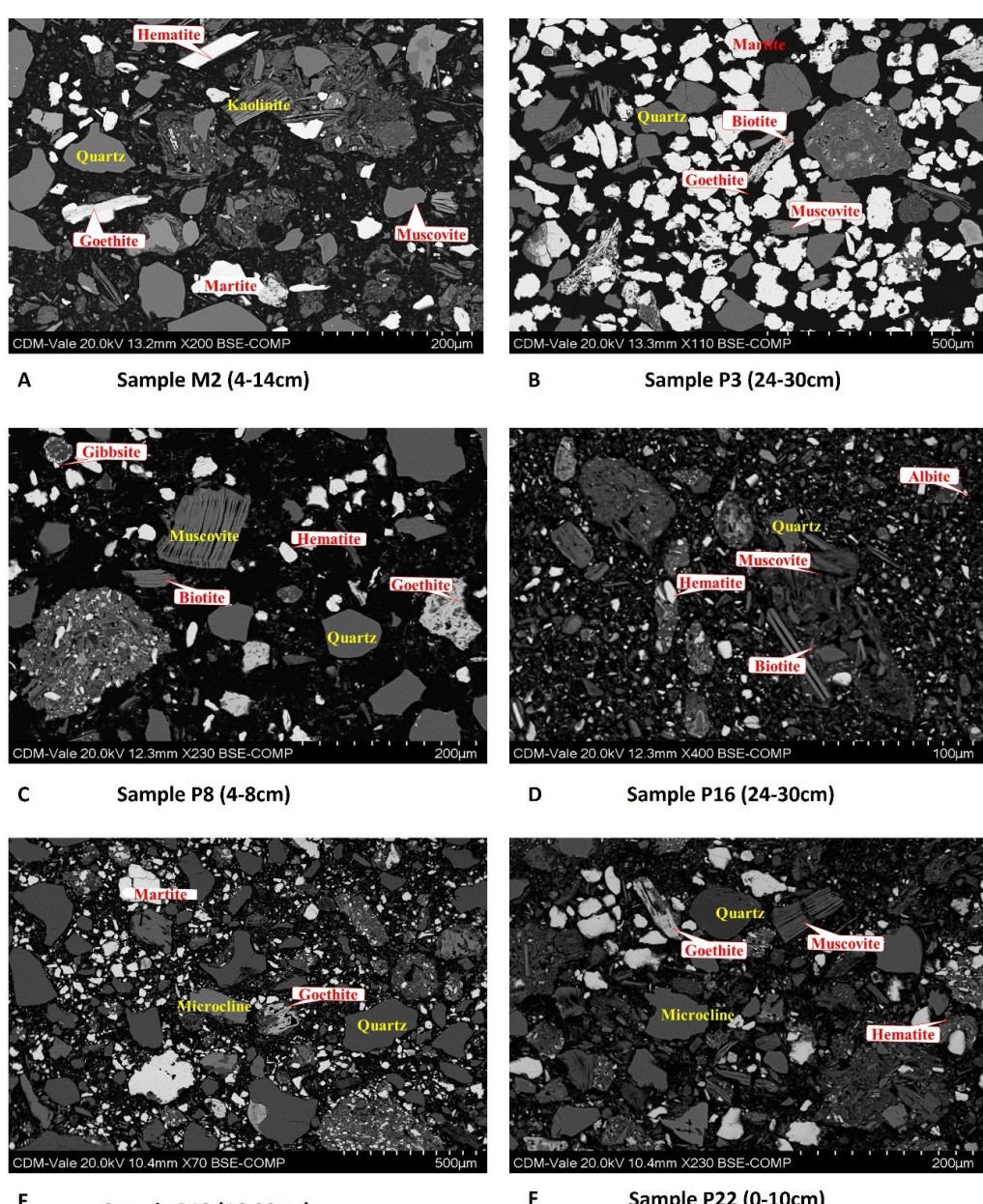

**Figure 4.** SEM BSE images of the analyzed borehole samples: (**A**) Sample M2, (**B**) Sample P3, (**C**) Sample P8, (**D**) Sample P16, (**E**) Sample P19 and (**F**) Sample P22.

The SEM images of samples P16-03 and P19-02 (Figure 4D,E) are characterized by the greater presence of biotite, quartz and muscovite in relation to iron oxides. In P22-01 (Figure 4F) there is an increase in particle size/granulometry and a significant reduction in the presence of iron oxides.

*4.1. M2*

The borehole M2 was located less than 1 km upstream of the Ferro-Carvão creek mouth. The core is 192 cm in length. The first 80 cm are characterized mainly by silt-clay

fragments with low opaque mineral concentration. The general aspect of the log suggests the occurrence of small fining upward cycles, which agrees with channel sedimentation (Figure 5).

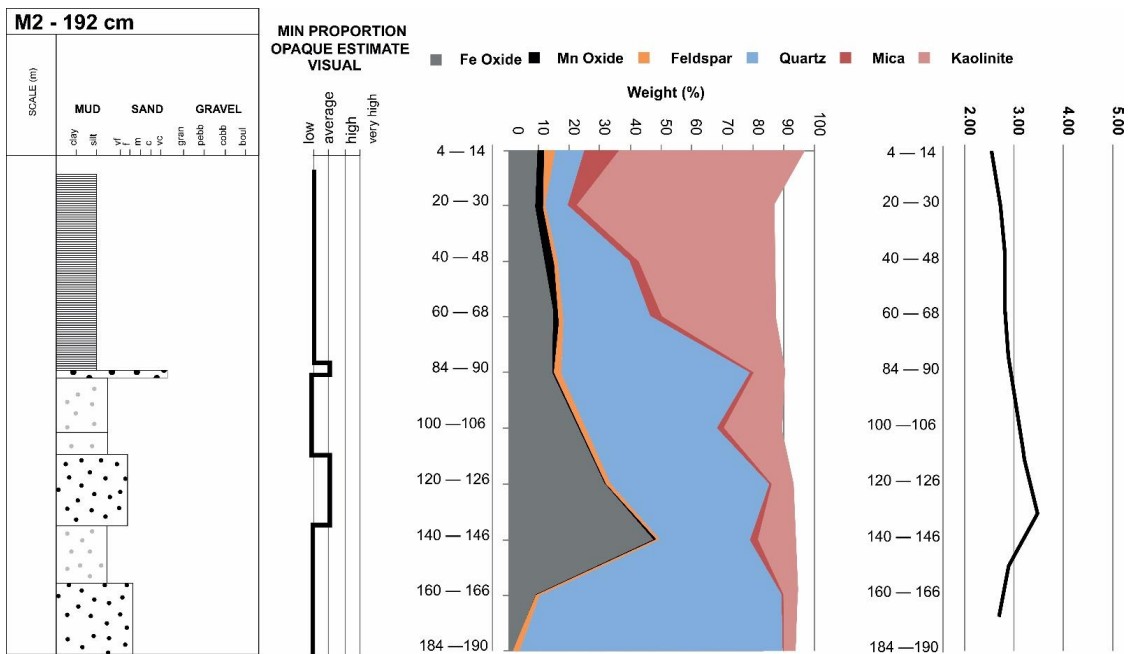

**Figure 5.** Multiproxy display of results obtained from M2 samples. From left to right one can observe the log with textural description, opaque relative proportions, modal mineralogy and density. Sampled intervals are indicated by the horizontal axis on the last two charts.

Kaolinite (>70%) is the predominant mineral in the top fine layer. Quartz content increases with depth together with iron oxide that reaches >40% in the 140–146 cm sample (Figure 4). Most of the density values range from 2.5 to 2.9, but they correlated with Fe oxide content and density values up to 3 are observed. Data from M2 is considered to be a reference for natural (pre-dam failure) sediments in the Paraopeba watershed.

### 4.2. P3

The P3 borehole is located approximately 4 km downstream from the tailings entrance point in the Paraopeba river. The core is 137 cm in length, consisting of laminae of very fine yellow sand and dark silt, with high opaque mineral content (Figure 6). These minerals display magnetism in variable degrees, and convoluted bedding was noted in top layers (see Figure 3b).

Minerals found in P3 samples are mainly iron oxides (>70%) and quartz (>10%) reflecting the laminations observed along the whole core. Accessories include Kaolinite and minerals from the mica group (Figure 4B). Density values are high, typically around 40 g/cm$^3$ over the core length (Figure 6).

### 4.3. P8

The P8 borehole is located approximately 10 km downstream from the confluence between Ferro Carvão creek and the Paraopeba river. The core is 156 cm in length. In the first meter (top to bottom), the silt-clay fraction is predominant consisting as an unstratified mud with high opaque mineral content (Figure 3c). This layer is mainly composed of Fe oxides with density values ranging upwards to 3.5 (Figure 7). In deeper levels, small fining upward cycles are observed, and quartz becomes the predominant mineral.

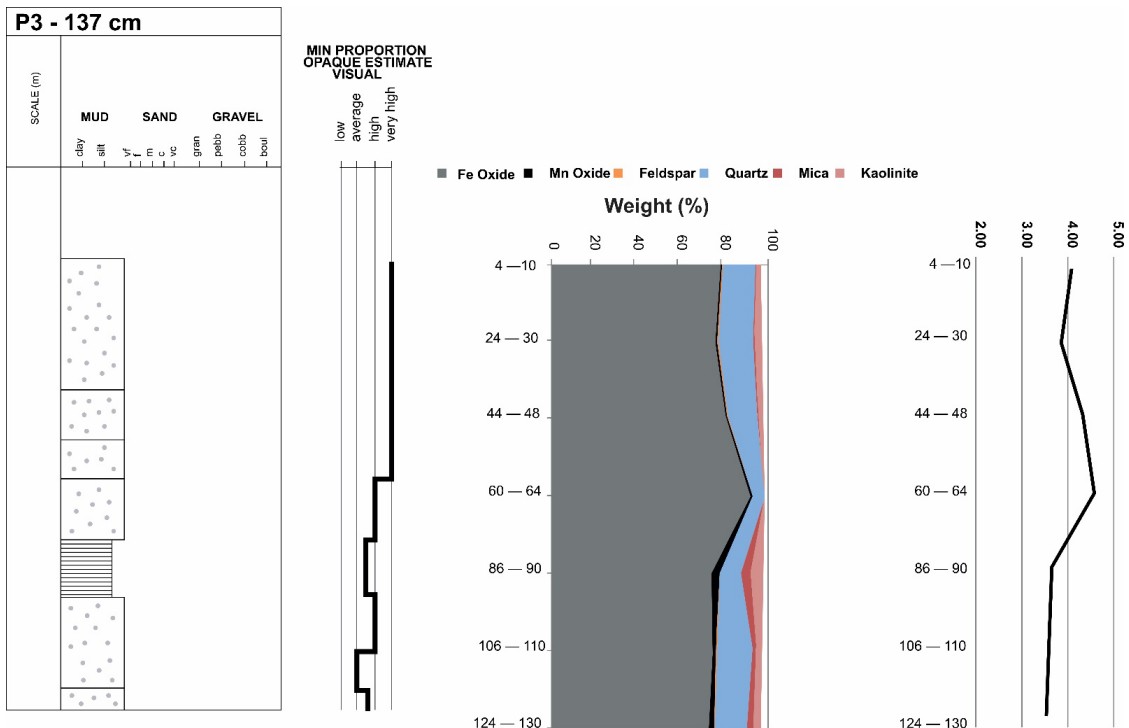

**Figure 6.** Multiproxy display of results obtained from P3 samples. From left to right one can observe the log with textural description, opaque relative proportions, modal mineralogy and density. Sampled intervals are indicated by the horizontal axis on the last two charts.

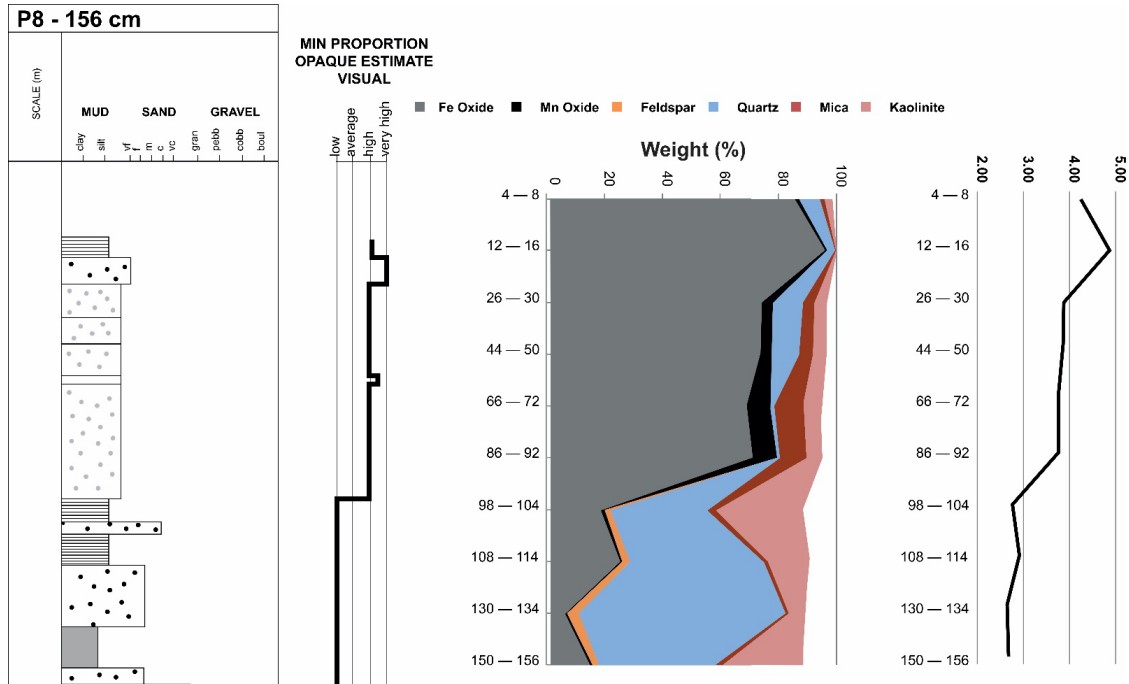

**Figure 7.** Multiproxy display of results obtained from P8 samples. From left to right one can observe the log with textural description, opaque relative proportions, modal mineralogy and density. Sampled intervals are indicated by the horizontal axis on the last two charts.

As observed in M2 samples, density values follow the iron and manganese oxide content. An abrupt difference in density values were measured between the 86–92 sample

and the 98–104 sample. Top layers show values higher than 3.5 g/cm$^3$, whereas bottom layers have density values below 3 g/cm$^3$.

### 4.4. P16

The P16 borehole was drilled approximately 20 km downstream from the tailings entrance point into the Paraopeba river. The core is 128 cm long and is predominantly composed of silt grains (Figure 8). The first 30 cm (top to bottom) shows a dark brown color reflecting a high opaque mineral content. Below this layer, sediment color switches to gray and then to dark yellow at the bottom of the profile.

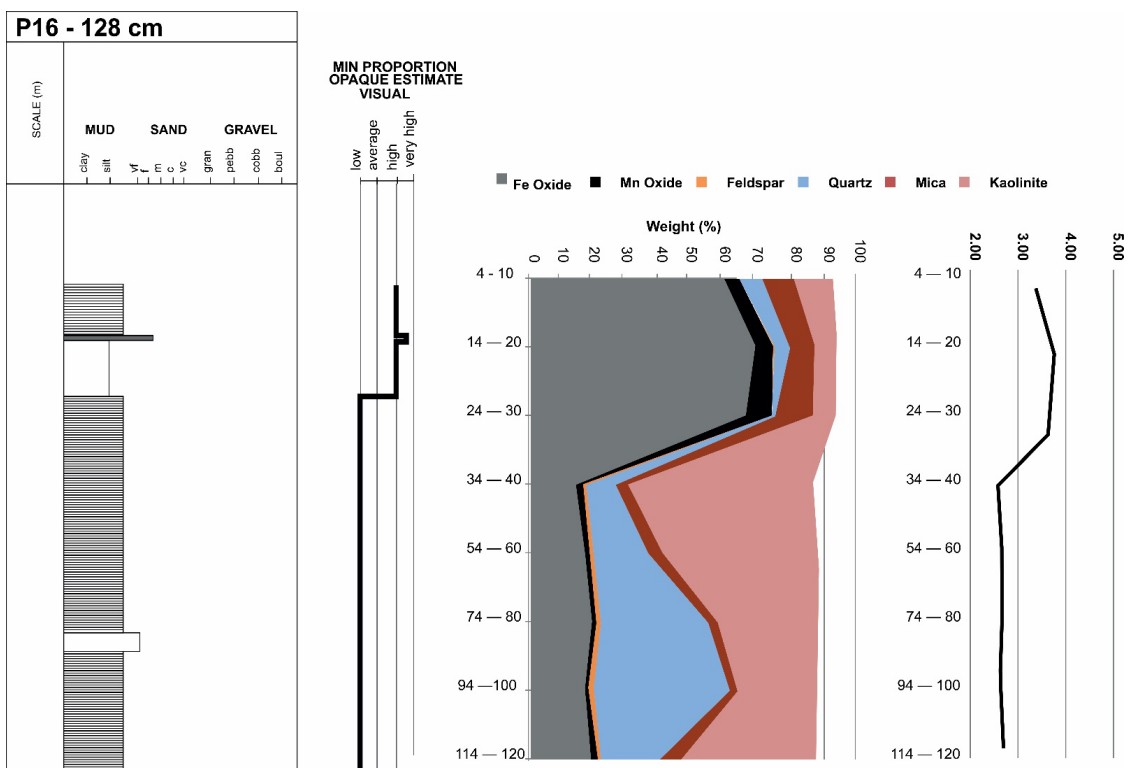

**Figure 8.** Multiproxy display of results obtained from P16 samples. From left to right one can observe the log with textural description, opaque relative proportions, modal mineralogy and density. Sampled intervals are indicated by the horizontal axis on the last two charts.

Together with color, abrupt changes on modal mineral content and density values are observed. The top brown layer presents Fe and Mn oxide concentration up to 70%, with additional quartz, mica and kaolinite (Figure 8). Deeper sections show a predominance of quartz and kaolinite with density values below 3 g/cm$^3$.

### 4.5. P19

The P19 borehole is located approximately 30 km downstream from the confluence of the Ferro do Carvão creek and Paraopeba river. The core obtained reached 116 cm in length (Figure 9). Opaque minerals are present in medium to high concentrations up to 58 cm deep (top to bottom), mostly found among fine sand and silt grains (Figure 3e). Below 60 cm depth, there is a fine layer composed of medium to fine sand grains (~80 cm) in between mud layers. Despite the grain size reduction, high concentrations of opaque minerals are not observed.

### 4.6. P22

The P22 borehole is located almost 40 km from the tailings entrance point in the Paraopeba river. The retrieved core is 158 cm long (Figure 10). Mud deposits can be

observed along the entire profile, sometimes interlayered with fine beds of silt and fine sand. Opaque mineral concentration varies from high to low in the first 20 cm deep (Figure 3f) and is predominantly low along the remaining sample length.

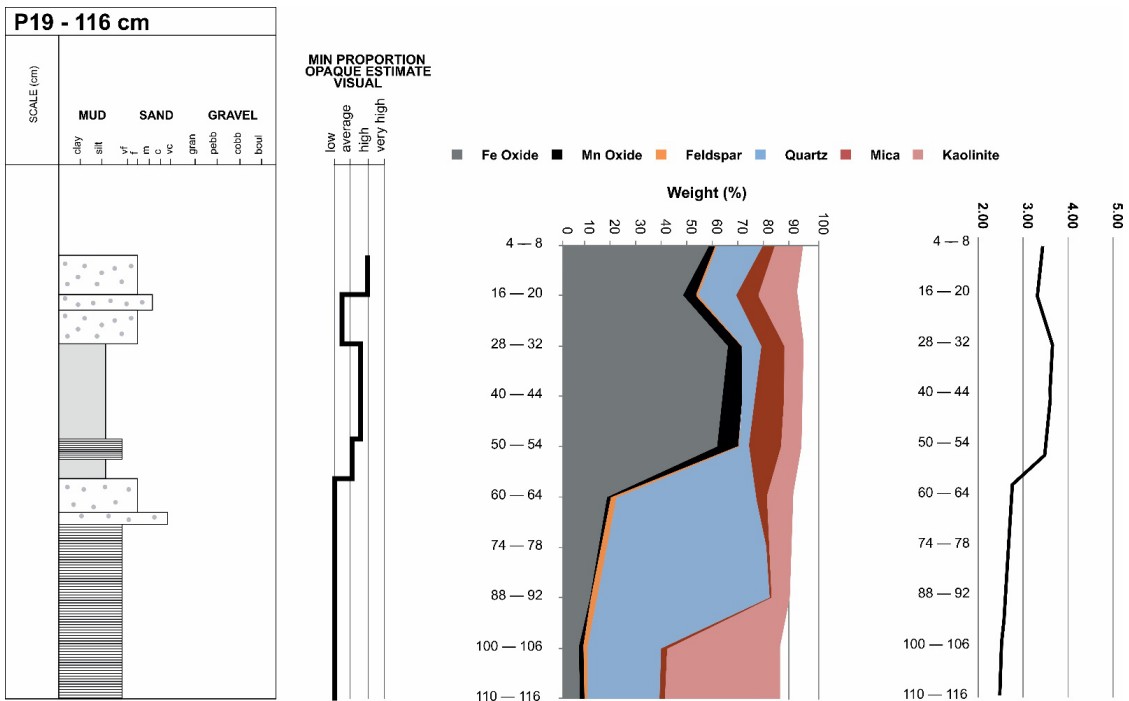

**Figure 9.** Multiproxy display of results obtained from P19 samples. From left to right one can observe the log with textural description, opaque relative proportions, modal mineralogy and density. Sampled intervals are indicated by the horizontal axis on the last two charts.

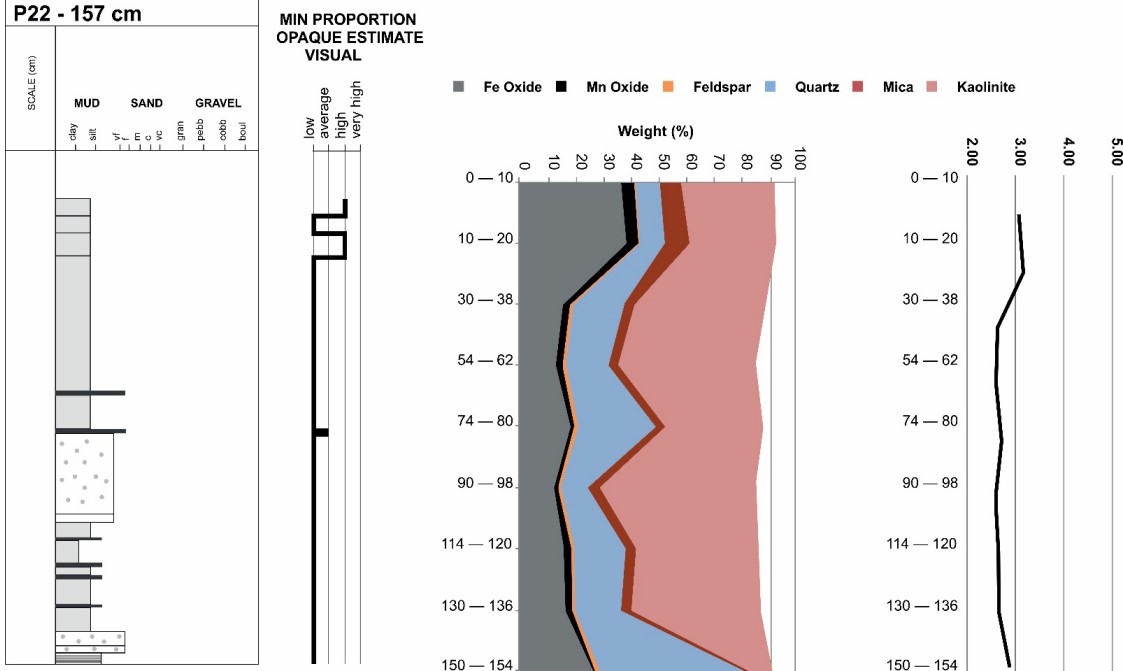

**Figure 10.** Multiproxy display of results obtained from P22 samples. From left to right one can observe the log with textural description, opaque relative proportions, modal mineralogy and density. Sampled intervals are indicated by the horizontal axis on the last two charts.

Only the two first samples (top to the bottom) showed a small increase in Fe and Mn oxide content (around 40%), reflected in density values that are a bit higher than 3 g/cm$^3$. Iron oxide concentration drops with increasing depth, as quartz and kaolinite distribution rise (Figure 10).

### 4.7. Mineralogical Data

Mineralogical images obtained by QEMSCAN (Figure 11) confirm patterns in the data previously described, with higher iron and manganese oxide concentrations (brownish color) observed in shallower portions of cores P3, P8, P16 and P19. The mineral association of natural (pre-failure) sediments can be assumed from core M2 where kaolinite, quartz and muscovite are predominant, but plagioclase, microcline and iron oxide may be observed in deeper portions. Similar mineral assemblages can be observed below the top oxide-rich layers in P3, P8, P16 and P19 (yellow-reddish colors).

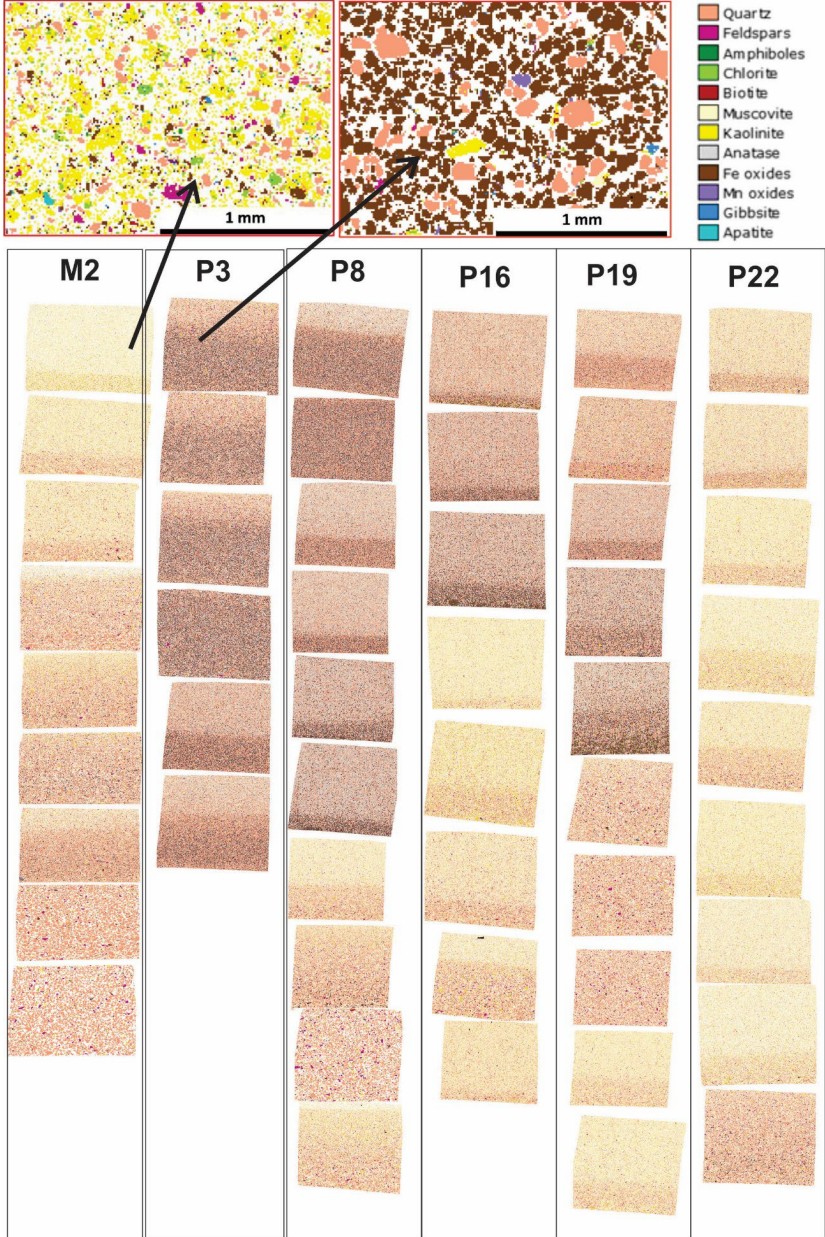

**Figure 11.** Comparative chart of mineralogical images obtained by QEMSCAN measurements of the sediments samples.

### 5. Discussion and Interpretations

Hierarchy cluster analyses (HCA) and principal components analyses (PCA) were applied to determine the main mineralogy shifts in the studied samples. Six groups were identified with an increase or decrease of one mineral phase or another (Figure 12). Quartz and kaolinite groups are explained by their stability through the weathering process and can be derived from most of the outcropping rocks occurring over the catchment. The hematite group, however, may be considered as an anomaly considering the large number of samples that fit into this group. Banded iron formations are found in adjacent ranges (Figure 1) and hematite is an important mineral phase in sediments of 1st to 3rd order streams that drain these ranges [33]. However, we maintain that the number of samples in the hematite group is too large, considering the extension of the drained area upstream of the study, the diversity of rock types found through this area and the total load of the Paraopeba River [44]. Furthermore, the abundance of samples within the hematite group has a direct association with tailings mineral composition [36]. Therefore, the presence of hematite in PCA analyses is assumed to be indicative of tailings deposition. This association is strengthened when PCA results considered together with core sample mineralogy results showing that Fe and Mn oxide minerals including hematite occur preferentially on the top section of the cores (Figure 13) and is not present in core M2, above the point of tailings entry into the Paraopeba River.

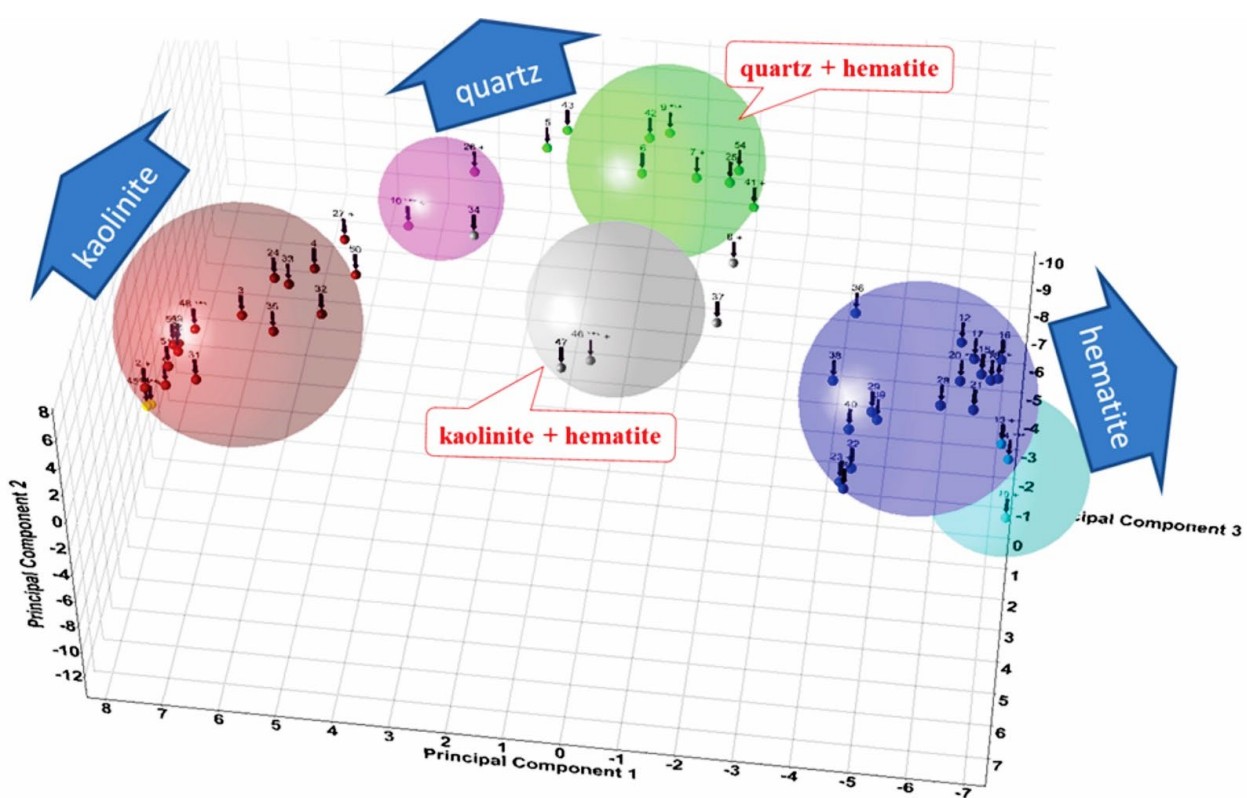

**Figure 12.** Principal components analysis—PCA for sample mineralogy.

Natural sediments from Core M2 and basal sediments in the remaining cores are distinguished by the following mineral associations: kaolinite, quartz + hematite, quartz and kaolinite (higher concentration). Hematite-bearing strata occur sporadically and punctually. Naturally occurring sediment has a density variation from 2.5 to 3.3 g/cm$^3$, with denser sediments occurring where higher iron oxide content is observed (Figure 13).

| Sample | Deph (cm) | Modal Composition/ True Density |
|--------|-----------|-------------------------------|
| M2 - 1 | 4 - 14 | 2,57 |
| M2 - 2 | 20 - 30 | 2,69 |
| M2 - 3 | 40 - 48 | 2,77 |
| M2 - 4 | 60 - 68 | 2,77 |
| M2 - 5 | 84 - 90 | 2,82 |
| M2 - 6 | 100 - 106 | 2,95 |
| M2 - 7 | 120 - 126 | 3,05 |
| M2 - 8 | 140 - 146 | 3,23 |
| M2 - 9 | 160 - 166 | 2,81 |
| M2 - 10 | 184 - 190 | 2,68 |

| Sample | Deph (cm) | Modal Composition/ True Density |
|--------|-----------|-------------------------------|
| P3 - 1 | 4 - 14 | 4,04 |
| P3 - 2 | 20 - 30 | 3,89 |
| P3 - 3 | 40 - 48 | 4,21 |
| P3 - 4 | 60 - 68 | 4,36 |
| P3 - 5 | 84 - 90 | 3,78 |
| P3 - 6 | 100 - 106 | 3,75 |
| P3 - 7 | 120 - 126 | 3,69 |

| Sample | Deph (cm) | Modal Composition/ True Density |
|--------|-----------|-------------------------------|
| P8 - 1 | 4 - 8 | 4,13 |
| P8 - 2 | 12 - 16 | 4,67 |
| P8 - 3 | 26 - 30 | 3,80 |
| P8 - 4 | 44 - 50 | 3,76 |
| P8 - 5 | 66 - 72 | 3,67 |
| P8 - 6 | 86 - 92 | 3,68 |
| P8 - 7 | 98 - 104 | 2,81 |
| P8 - 8 | 108 - 114 | 2,93 |
| P8 - 9 | 130 - 134 | 2,71 |
| P8 - 10 | 150 - 156 | 2,73 |

| Sample | Deph (cm) | Modal Composition/ True Density |
|--------|-----------|-------------------------------|
| P16 - 1 | 4 - 10 | 3,54 |
| P16 - 2 | 14 - 20 | 3,73 |
| P16 - 3 | 24 - 30 | 3,63 |
| P16 - 4 | 34 - 40 | 2,79 |
| P16 - 5 | 54 - 60 | 2,84 |
| P16 - 6 | 74 - 80 | 2,84 |
| P16 - 7 | 94 - 100 | 2,82 |
| P16 - 8 | 114 - 120 | 2,87 |

| Sample | Deph (cm) | Modal Composition/ True Density |
|--------|-----------|-------------------------------|
| P19 - 1 | 4 - 8 | 3,41 |
| P19 - 2 | 16 - 20 | 3,32 |
| P19 - 3 | 28 - 32 | 3,65 |
| P19 - 4 | 40 - 44 | 3,56 |
| P19 - 5 | 50 - 54 | 3,50 |
| P19 - 6 | 60 - 64 | 2,88 |
| P19 - 7 | 74 - 78 | 2,81 |
| P19 - 8 | 88 - 92 | 2,77 |
| P19 - 9 | 100 - 106 | 2,71 |
| P19 - 10 | 110 - 116 | 2,71 |

| Sample | Deph (cm) | Modal Composition/ True Density |
|--------|-----------|-------------------------------|
| P22 - 1 | 0 - 10 | 3,05 |
| P22 - 2 | 10 - 20 | 3,11 |
| P22 - 3 | 30 - 38 | 2,80 |
| P22 - 4 | 54 - 62 | 2,79 |
| P22 - 5 | 74 - 80 | 2,85 |
| P22 - 6 | 90 - 98 | 2,78 |
| P22 - 7 | 114 - 120 | 2,81 |
| P22 - 8 | 130 - 136 | 2,82 |
| P22 - 9 | 150 - 154 | 2,93 |

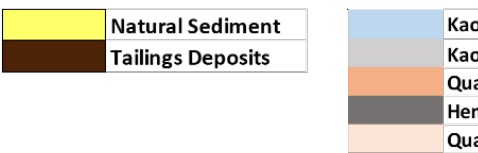

- Natural Sediment
- Tailings Deposits

- Kaolinite
- Kaolinite + Hematite
- Quartz + Hematite
- Hematite
- Quartz
- Hematite (high concentration)

**Figure 13.** Principal component analysis (PCA) of natural sediments and tailings.

Tailings deposits on the other hand, are characterized by high-density values, above 3.5 g/cm$^3$, with predominant associations of hematite, hematite (high concentration) and kaolinite + hematite.

Rivers are dynamic environments. They transport a natural load of nutrients and sediment, but they still receive and transport a large amount of human waste. According to Lima et al. (2020) [45] tailings derived from B1 Dam burst entered the Paraopeba river as a low energy mud flow. Very fine particles were transported as a wash load (turbidity plume) and moved farther downstream through the thalweg. Fine and course particles have been transported as suspended and bedload as the Paraopeba River attempted to find a new dynamic equilibrium between erosion and deposition.

Our results suggest a decrease in the thickness of the tailings layers along the main channel of the river, with around 20 cm of tailings being observed in P22 (Figure 13), the region closest to the Igarapé Thermoelectric Plant reservoir (Figure 14). Considering the depositional characteristic provided by the reservoir, due to the change from lotic to the lentic environment, a significant reduction in the sediment transport downstream of such structure is expected. Nevertheless, this is not expected to be a permanent condition, rather a snapshot in time of a condition that is constantly evolving.

Studies performed prior to the failure of the B1 Dam describe the presence of metals and other potentially toxic elements (Fe, Mn, As, Cr, and Ni) in high concentrations in the stream sediments from the Paraopeba River basin (i.e., [15,35,46]). These studies suggest these elements are possibly naturally occurring in some regions. Despite the great relevance of geochemical studies in environmental management, assessing the provenance of contaminants in such a complex scenario will require a previous distinction between whether the contaminant is associated to tailing deposits or natural sediments. Our results show that mineralogical associations are a key element to this provenance assessment.

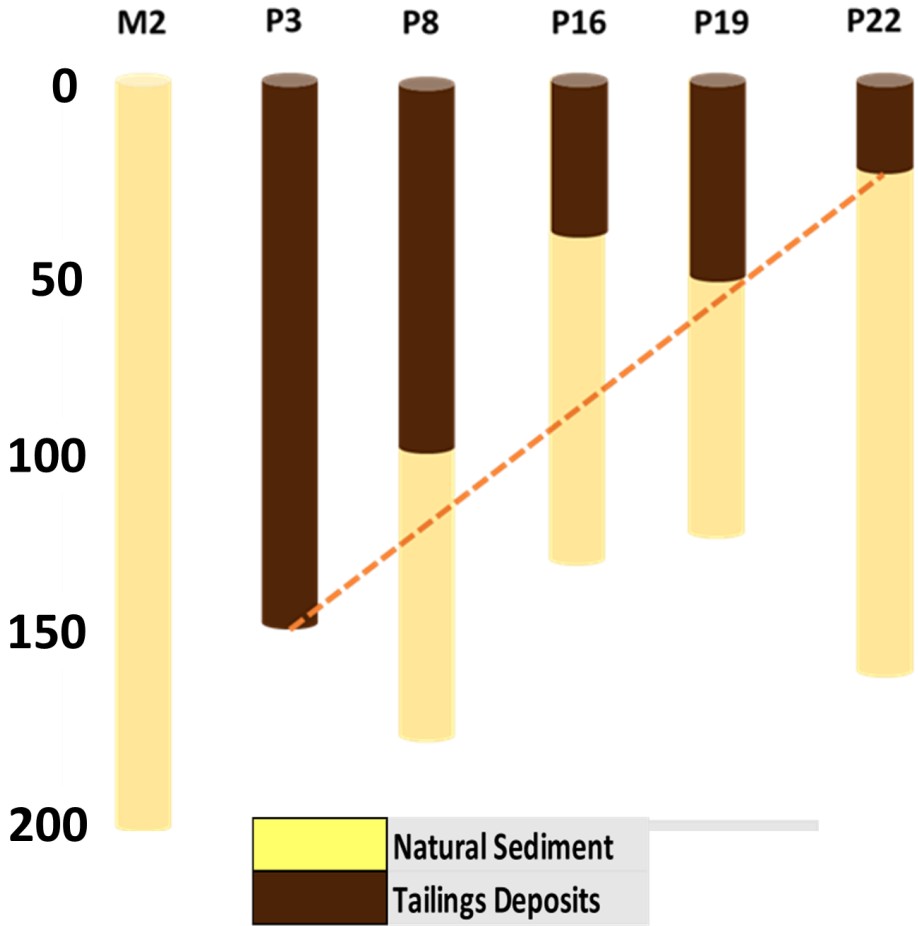

**Figure 14.** Vertical distribution (cm) of sediments and tailings along the Paraopeba River.

## 6. Conclusions

Paraopeba River received an additional load of 1.6 Mm$^3$ of iron ore tailings following the failure of B1 Dam. Eight months after the accident, the tailings could be recognized as different sedimentary facies occurring in the highest layers of the channel bedload. PCA analysis of SEM-based mineralogy data identified six different mineralogical groups, characterized by different contents of quartz, kaolinite and hematite. Natural (pre-failure) sediments may be composed of different proportions of all groups, but preferentially those related to quartz and kaolinite. Tailings are mainly composed by groups dominated by hematite.

Because potentially toxic elements were available in the Paraopeba river sediments before the B1 Dam failure, detailed geochemical provenance investigations will be required. Our data suggests that mineralogically there is a great change in the record and that can be used to tracing tailings downstream. Mineralogical oriented studies coupled to geochemical can enhance understanding of the source and origin of geochemical anomalies, if from the long-time regular deposits or from samples which can be attributed to the dam burst without reasonable doubt. In this sense, SEM-based mineralogy has an enormous potential in environment studies.

**Author Contributions:** Conceptualization, F.V.L. and R.K.-R.; methodology, F.V.L., R.K.-R. and L.G.; writing—original draft preparation, F.V.L., R.K.-R. and L.P.L.; writing—review and editing, F.V.L., R.K.-R., L.G. and L.P.L. All authors have read and agreed to the published version of the manuscript.

**Funding:** This research received no external funding.

**Data Availability Statement:** Not applicable.

**Acknowledgments:** The authors wish to express their deep gratitude to Fred Ford (Vale Canada) for the detailed and patient revision of either lexical and technical aspects of the manuscript.

**Conflicts of Interest:** The authors declare no conflict of interest.

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
