# Peer review of "Mineralogical Fingerprint of Iron Ore Tailings in Paraopeba River Bedload Sediments after the B1 Dam Failure in Brumadinho, MG (Brazil)"

_minerals, doi:10.3390/min12060716_

Round 1
Reviewer 1 Report
The paper is dedicated to SEM mineralogical analysis to distinguish between natural sediments and tailings from iron ore ponds. The topic is relevant and falls within the scope of Minerals magazine. The paper contributes to the identification of the best solutions in case of such ecological accidents.
The introduction clearly sets out the issue under investigation. The methodology presented, the methods and the investigations performed are adequate. The procedures are arranged in a logical way. The method used and the results obtained are explained in detail for each sample analyzed. The conclusions are supported by the results.
Author Response
Dear Reviewer, thank you for your contributions! They were certainly very enriching for our work.
We try to answer them in the best possible way.
We hope they are as requested.
With regards.

Reviewer 2 Report
This is another paper adding scientifically sound data to help to evaluate the true effect of the Brumadinho Dam burst in January 2019. Here the mineralogy of five cores of sediments potentially affected by the accident is compared to one core upstream, thus not-affected.
The methods are adequately described, and the results are simple but efficient for the proposed objective, starting to define a 3D distribution of the effects of the dam burst form a mineralogical point of view.
As for the proposed method the manuscript succeeds in proving it feasible and adequate. Certainly, more sampling points are necessary to get a better-resolution 3D map, and it was not aimed at.
But the manuscript also mentions (more than once) potentially toxic elements, although no chemical analysis was presented. The relation of the presented data and toxic elements geochemistry is hinted at in the conclusion, suggesting repeating the performed work when addressing it, to which I agree. But it sounds very much like a rebuttal to the spread of toxic elements due to the dam burst - and this without any analysis to prove it, than actually a logic conclusion. Considering most of the authors work for the company responsible for the failed dam, there could be suspicion of competing interests at this final, unverified, conclusion, which would compromise the integrity of the whole study. The mineralogical analyses as presented, coupled to chemical assays for the same samples, could prove the point of this last conclusion, or not. Alternatively, the mentioned environmental or geochemical “jeopardy” (in abstract and conclusions) should give way just to the proved conclusion that mineralogically there is a great change in the record, and that further analyses must also take into account where exactly the samples come from, if from the long-time regular deposits or from samples which can be attributed to the dam burst without reasonable doubt. And that is the reason for the paper, suggesting how to do it.
Mineral names do not require start with caption.
Author Response

(The authors gave the same response as above.)

Reviewer 3 Report
Dear Editor,
I read the submitted work, I think it's a pretty interesting work and I think it's written and well presented. For these reasons, the work is suitable for publication. I don't have particular comments or suggestions to make, but I noticed some inaccuracies regarding some figures. I think that the figures are very important to understand the work and to observe the results, in this case i think some figures should be revised and improved. In any case all comments and suggestions are in the attached file. I suggest the authors to improve the quality of several images because, in asome cases, it is not very easy to understand and read the text within.
I hope my comments can help you to improve the ms quality

Author Response

(The authors gave the same response as above.)
